# Learning from Trajectories via Subgoal Discovery

**Sujoy Paul**[1]
supaul@ece.ucr.edu

**Jeroen van Baar**[2]
jeroen@merl.com

**Amit K. Roy-Chowdhury**[1]
amitrc@ece.ucr.edu

[1]University of California-Riverside    [2]Mitsubishi Electric Research Laboratories (MERL)

## Abstract

Learning to solve complex goal-oriented tasks with sparse terminal-only rewards often requires an enormous number of samples. In such cases, using a set of expert trajectories could help to learn faster. However, Imitation Learning (IL) via supervised pre-training with these trajectories may not perform as well and generally requires additional finetuning with expert-in-the-loop. In this paper, we propose an approach which uses the expert trajectories and learns to decompose the complex main task into smaller sub-goals. We learn a function which partitions the state-space into sub-goals, which can then be used to design an extrinsic reward function. We follow a strategy where the agent first learns from the trajectories using IL and then switches to Reinforcement Learning (RL) using the identified sub-goals, to alleviate the errors in the IL step. To deal with states which are under-represented by the trajectory set, we also learn a function to modulate the sub-goal predictions. We show that our method is able to solve complex goal-oriented tasks, which other RL, IL or their combinations in literature are not able to solve.

## 1   Introduction

Reinforcement Learning (RL) aims to take sequential actions so as to maximize, by interacting with an environment, a certain pre-specified reward function, designed for the purpose of solving a task. RL using Deep Neural Networks (DNNs) has shown tremendous success in several tasks such as playing games [1, 2], solving complex robotics tasks [3, 4], etc. However, with sparse rewards, these algorithms often require a huge number of interactions with the environment, which is costly in real-world applications such as self-driving cars [5], and manipulations using real robots [3]. Manually designed dense reward functions could mitigate such issues, however, in general, it is difficult to design detailed reward functions for complex real-world tasks.

Imitation Learning (IL) using trajectories generated by an expert can potentially be used to learn the policies faster [6]. But, the performance of IL algorithms [7] are not only dependent on the performance of the expert providing the trajectories, but also on the state-space distribution represented by the trajectories, especially in case of high dimensional states. In order to avoid such dependencies on the expert, some methods proposed in the literature [8, 9] take the path of combining RL and IL. However, these methods assume access to the expert value function, which may become impractical in real-world scenarios.

In this paper, we follow a strategy which starts with IL and then switches to RL. In the IL step, our framework performs supervised pre-training which aims at learning a policy which best describes the expert trajectories. However, due to limited availability of expert trajectories, the policy trained with IL will have errors, which can then be alleviated using RL. Similar approaches are taken in [9] and [10], where the authors show that supervised pre-training does help to speed-up learning. However, note that the reward function in RL is still sparse, making it difficult to learn. With this in mind, we

pose the following question: *can we make more efficient use of the expert trajectories, instead of just supervised pre-training?*

Given a set of trajectories, humans can quickly identify waypoints, which need to be completed in order to achieve the goal. We tend to break down the entire complex task into sub-goals and try to achieve them in the best order possible. Prior knowledge of humans helps to achieve tasks much faster [11, 12] than using only the trajectories for learning. The human psychology of divide-and-conquer has been crucial in several applications and it serves as a motivation behind our algorithm which learns to partition the state-space into sub-goals using expert trajectories. The learned sub-goals provide a discrete reward signal, unlike value based continuous reward [13, 14], which can be erroneous, especially with a limited number of trajectories in long time horizon tasks. As the expert trajectories set may not contain all the states where the agent may visit during exploration in the RL step, we augment the sub-goal predictor via one-class classification to deal with such under-represented states. We perform experiments on three goal-oriented tasks on MuJoCo [15] with sparse terminal-only reward, which state-of-the-art RL, IL or their combinations are not able to solve.

## 2 Related Works

Our work is closely related to learning from demonstrations or expert trajectories as well as discovering sub-goals in complex tasks. We first discuss works on imitation learning using expert trajectories or reward-to-go. We also discuss the methods which aim to discover sub-goals, in an online manner during the RL stage from its past experience.

**Imitation Learning.** Imitation Learning [16, 17, 18, 19, 20] uses a set of expert trajectories or demonstrations to guide the policy learning process. A naive approach to use such trajectories is to train a policy in a supervised learning manner. However, such a policy would probably produce errors which grow quadratically with increasing steps. This can be alleviated using Behavioral Cloning (BC) algorithms [7, 21, 22], which queries expert action at states visited by the agent, after the initial supervised learning phase. However, such query actions may be costly or difficult to obtain in many applications. Trajectories are also used by [23], to guide the policy search, with the main goal of optimizing the return of the policy rather than mimicking the expert. Recently, some works [8, 24, 14] aim to combine IL with RL by assuming access to experts reward-to-go at the states visited by the RL agent. [9] take a moderately different approach where they switch from IL to RL and show that randomizing the switch point can help to learn faster. The authors in [25] use demonstration trajectories to perform skill segmentation in an Inverse Reinforcement Learning (IRL) framework. The authors in [26] also perform expert trajectory segmentation, but do not show results on learning the task, which is our main goal. SWIRL [27] make certain assumptions on the expert trajectories to learn the reward function and their method is dependent on the discriminability of the state features, which we on the other hand learn end-to-end.

**Learning with Options.** Discovering and learning options have been studied in the literature [28, 29, 30] which can be used to speed-up the policy learning process. [31] developed a framework for planning based on options in a hierarchical manner, such that low level options can be used to build higher level options. [32] propose to learn a set of options, or skills, by augmenting the state space with a latent categorical skill vector. A separate network is then trained to learn a policy over options. The Option-Critic architecture [33] developed a gradient based framework to learn the options along with learning the policy. This framework is extended in [34] to handle a hierarchy of options. [35] proposed a framework where the goals are generated using Generative Adversarial Networks (GAN) in a curriculum learning manner with increasingly difficult goals. Researchers have shown that an important way of identifying sub-goals in several tasks is identifying bottle-neck regions in tasks. Diverse Density [36], Relative Novelty [37], Graph Partitioning [38], clustering [39] can be used to identify such sub-goals. However, unlike our method, these algorithms do not use a set of expert trajectories, and thus would still be difficult to identify useful sub-goals for complex tasks.

## 3 Methodology

We first provide a formal definition of the problem we are addressing in this paper, followed by a brief overall methodology, and then present a detailed description of our framework.

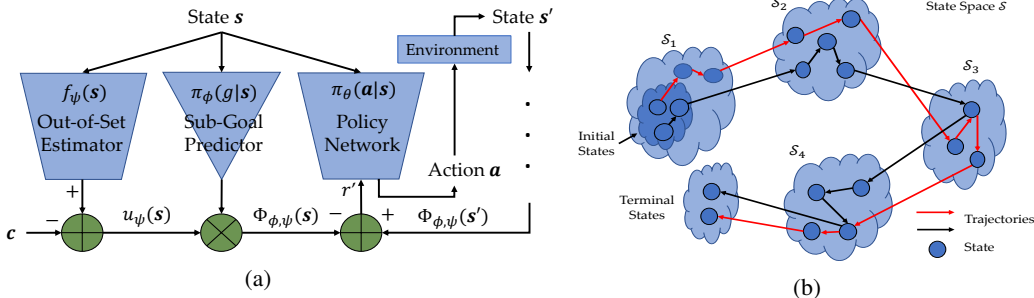

Figure 1: (a) This shows an overview of our proposed framework to train the policy network along with sub-goal based reward function with out-of-set augmentation. (b) An example state-partition with two independent trajectories in black and red. Note that the terminal state is shown as a separate state partition because we assume it to be indicated by the environment and not learned.

**Problem Definition.** Consider a standard RL setting where an agent interacts with an environment which can be modeled by a Markov Decision Process (MDP) $\mathcal{M} = (\mathcal{S}, \mathcal{A}, \mathcal{P}, r, \gamma, \mathcal{P}_0)$, where $\mathcal{S}$ is the set of states, $\mathcal{A}$ is the set of actions, $r$ is a scalar reward function, $\gamma \in [0, 1]$ is the discount factor and $\mathcal{P}_0$ is the initial state distribution. Our goal is to learn a policy $\pi_\theta(\boldsymbol{a}|\boldsymbol{s})$, with $\boldsymbol{a} \in \mathcal{A}$, which optimizes the expected discounted reward $\mathbb{E}_\tau[\sum_{t=0}^{\infty} \gamma^t r(\boldsymbol{s}_t, \boldsymbol{a}_t)]$, where $\tau = (\dots, \boldsymbol{s}_t, \boldsymbol{a}_t, r_t, \dots)$ and $\boldsymbol{s}_0 \sim \mathcal{P}_0$, $\boldsymbol{a}_t \sim \pi_\theta(\boldsymbol{a}|\boldsymbol{s}_t)$ and $\boldsymbol{s}_{t+1} \sim \mathcal{P}(\boldsymbol{s}_{t+1}|\boldsymbol{s}_t, a_t)$.

With sparse rewards, optimizing the expected discounted reward using RL may be difficult. In such cases, it may be beneficial to use a set of state-action trajectories $\mathcal{D} = \{\{(\boldsymbol{s}_{ti}, \boldsymbol{a}_{ti}^*)\}_{t=1}^{n_i}\}_{i=1}^{n_d}$ generated by an expert to guide the learning process. $n_d$ is the number of trajectories in the dataset and $n_i$ is the length of the $i^{th}$ trajectory. We propose a methodology to efficiently use $\mathcal{D}$ by discovering sub-goals from these trajectories and use them to develop an extrinsic reward function.

**Overall Methodology.** Several complex, goal-oriented, real-world tasks can often be broken down into sub-goals with some natural ordering. Providing positive rewards after completing these sub-goals can help to learn much faster compared to sparse, terminal-only rewards. In this paper, we advocate that such sub-goals can be learned directly from a set of expert demonstration trajectories, rather than manually designing them.

A pictorial description of our method is presented in Fig. 1a. We use the set $\mathcal{D}$ to first train a policy by applying supervised learning. This serves a good initial point for policy search using RL. However, with sparse rewards, the search can still be difficult and the network may forget the learned parameters in the first step if it does not receive sufficiently useful rewards. To avoid this, we use $\mathcal{D}$ to learn a function $\pi_\phi(g|\boldsymbol{s})$, which given a state, predicts sub-goals. We use this function to obtain a new reward function, which intuitively informs the RL agent whenever it moves from one sub-goal to another. We also learn a utility function $u_\psi(\boldsymbol{s})$ to modulate the sub-goal predictions over the states which are not well-represented in the set $\mathcal{D}$. We approximate the functions $\pi_\theta$, $\pi_\phi$, and $u_\psi$ using neural networks. We next describe our meaning of sub-goals followed by an algorithm to learn them.

### 3.1 Sub-goal Definition

**Definition 1.** Consider that the state-space $\mathcal{S}$ is partitioned into sets of states as $\{\mathcal{S}_1, \mathcal{S}_2, \dots, \mathcal{S}_{n_g}\}$, s.t., $\mathcal{S} = \cup_{i=1}^{n_g} \mathcal{S}_i$ and $\cap_{i=1}^{n_g} \mathcal{S}_i = \emptyset$ and $n_g$ is the number of sub-goals specified by the user. For each $(\boldsymbol{s}, \boldsymbol{a}, \boldsymbol{s}')$, we say that the particular action takes the agent from one sub-goal to another iff $\boldsymbol{s} \in \mathcal{S}_i$, $\boldsymbol{s}' \in \mathcal{S}_j$ for some $i, j \in G = \{1, 2, \dots, n_g\}$ and $i \neq j$.

We assume that there is an ordering in which groups of states appear in the trajectories as shown in Fig. 1b. However, the states within these groups of states may appear in any random order in the trajectories. These groups of states are not defined a priori and our algorithm aims at estimating these partitions. Note that such orderings are natural in several real-world applications where a certain sub-goal can only be reached after completing one or more previous sub-goals. We show (empirically in the supplementary) that our assumption is soft rather than being strict, i.e., the degree by which the trajectories deviate from the assumption determines the granularity of the discovered sub-goals. We may consider that states in the trajectories of $\mathcal{D}$ appear in increasing order of sub-goal indices,

i.e., achieving sub-goal $j$ is harder than achieving sub-goal $i$ $(i < j)$. This gives us a natural way of defining an extrinsic reward function, which would help towards faster policy search. Also, all the trajectories in $\mathcal{D}$ should start from the initial state distribution and end at the terminal states.

## 3.2  Learning Sub-Goal Prediction

We use $\mathcal{D}$ to partition the state-space into $n_g$ sub-goals, with $n_g$ being a hyperparameter. We learn a neural network to approximate $\pi_\phi(g|s)$, which given a state $s \in \mathcal{S}$ predicts a probability mass function (p.m.f.) over the possible sub-goal partitions $g \in G$. The order in which the sub-goals occur in the trajectories, i.e., $\mathcal{S}_1 < \mathcal{S}_2 < \cdots < \mathcal{S}_{n_g}$, acts as a supervisory signal, which can be derived from our assumption mentioned above.

We propose an iterative framework to learn $\pi_\phi(g|s)$ using these ordered constraints. In the first step, we learn a mapping from states to sub-goals using equipartition labels among the sub-goals. Then we infer the labels of the states in the trajectories and correct them by imposing ordering constraints. We use the new labels to again train the network and follow the same procedure until convergence. These two steps are as follows.

**Learning Step.** In this step we consider that we have a set of tuples $(s, g)$, which we use to learn the function $\pi_\phi$, which can be posed as a multi-class classification problem with $n_g$ categories. We optimize the following cross-entropy loss function,

$$\pi_\phi^* = \arg\min_{\pi_\phi} \frac{1}{N} \sum_{i=1}^{n_d} \sum_{t=1}^{n_i} \sum_{k=1}^{n_g} -\mathbf{1}\{g_{ti} = k\} \log \pi_\phi(g = j | s_{ti}) \tag{1}$$

where $\mathbf{1}$ is the indicator function and $N$ is the number of states in the dataset $\mathcal{D}$. To begin with, we do not have any labels $g$, and thus we consider equipartition of all the sub-goals in $G$ along each trajectory. That is, given a trajectory of states $\{s_{1i}, s_{2i}, \ldots, s_{n_i i}\}$ for some $i \in \{1, 2, \ldots, n_d\}$, the initial sub-goals are,

$$g_{ti} = j, \quad \forall \lfloor \frac{(j-1)n_i}{n_g} \rfloor < t <= \lfloor \frac{jn_i}{n_g} \rfloor, \ j \in G \tag{2}$$

Using this initial labeling scheme, similar states across trajectories may have different labels, but the network is expected to converge at the Maximum Likelihood Estimate (MLE) of the entire dataset. We also optimize CASL [40] for stable learning as the initial labels can be erroneous. In the next iteration of the learning step, we use the inferred sub-goal labels, which we obtain as follows.

**Inference Step.** Although the equipartition labels in Eqn. 2 may have similar states across different trajectories mapped to dissimilar sub-goals, the learned network modeling $\pi_\phi$ provides maps similar states to the same sub-goal. But, Eqn. 1, and thus the predictions of $\pi_\phi$ does not account for the natural temporal ordering of the sub-goals. Even with architectures such as Recurrent Neural Networks (RNN), it may be better to impose such temporal order constraints explicitly rather than relying on the network to learn them. We inject such order constraints using Dynamic Time Warping (DTW).

Formally, for the $i^{th}$ trajectory in $\mathcal{D}$, we obtain the following set: $\{(s_{ti}, \pi_\phi(g|s_{ti})\}_{t=1}^{n_i}$, where $\pi_\phi$ is a vector representing the p.m.f. over the sub-goals $G$. However, as the predictions do not consider temporal ordering, the constraint that sub-goal $j$ occurs after sub-goal $i$, for $i < j$, is not preserved. To impose such constraints, we use DTW between the two sequences $\{e_1, e_2, \ldots, e_{n_g}\}$, which are the standard basis vectors in the $n_g$ dimensional Euclidean space and $\{\pi_\phi(g|s_{1i}), \pi_\phi(g|s_{2i}), \ldots, \pi_\phi(g|s_{n_i i})\}$. We use the $l1$-norm of the difference between two vectors as the similarity measure in DTW. In this process, we obtain a sub-goal assignment for each state in the trajectories, which become the new labels for training in the learning step.

We then invoke the learning step using the new labels (instead of Eqn. 2), followed by the inference step to obtain the next sub-goal labels. We continue this process until the number of sub-goal labels changed between iterations is less than a certain threshold. This method is presented in Algorithm 1, where the superscript $k$ represents the iteration number in learning-inference alternates.

**Reward Using Sub-Goals.** The ordering of the sub-goals, as discussed before, provides a natural way of designing a reward function as follows:

$$r'(s, a, s') = \gamma * \arg\max_{j \in G} \pi_\phi(g = j | s') - \arg\max_{k \in G} \pi_\phi(g = k | s) \tag{3}$$

where the agent in state $\boldsymbol{s}$ takes action $a$ and reaches state $\boldsymbol{s}'$. The augmented reward function would become $r + r'$. Considering that we have a function of the form $\Phi_\phi(\boldsymbol{s}) = \arg\max_{j \in G} \pi_\phi(g = j|\boldsymbol{s})$, and without loss of generality that $G = \{0, 1, \dots, n_g - 1\}$, so that for the initial state $\Phi_\phi(\boldsymbol{s}_0) = 0$, it follows from [13] that every optimal policy in $\mathcal{M}' = (\mathcal{S}, \mathcal{A}, \mathcal{P}, r + r', \gamma, \mathcal{P}_0)$, will also be optimal in $\mathcal{M}$. However, the new reward function may help to learn the task faster.

**Out-of-Set Augmentation.** In several applications, it might be the case that the trajectories only cover a small subset of the state space, while the agent, during the RL step, may visit states outside of the states in $\mathcal{D}$. The sub-goals estimated at these out-of-set states may be erroneous. To alleviate this problem, we use a logical assertion on the potential function $\Phi_\phi(\boldsymbol{s})$ that the sub-goal predictor is confident only for states which are well-represented in $\mathcal{D}$, and not elsewhere. We learn a neural network to model a utility function $u_\psi : \mathcal{S} \to \mathbb{R}$, which given a state, predicts the degree by which it is seen in the dataset $\mathcal{D}$. To do this, we build upon Deep One-Class Classification [41], which performs well on the task of anomaly detection. The idea is derived from Support Vector Data Description (SVDD) [42], which aims to find the smallest hypersphere enclosing the given data points with minimum error. Data points outside the sphere are then deemed as anomalous. We learn the parameters of $u_\psi$ by optimizing the following function:

$$\psi^* = \arg\min_\psi \frac{1}{N} \sum_{i=1}^{n_d} \sum_{t=1}^{n_i} ||f_\psi(\boldsymbol{s}_{ti}) - \boldsymbol{c}||^2 + \lambda||\psi||_2^2,$$

where $\boldsymbol{c} \in \mathbb{R}^m$ is a vector determined a priori [41], $f$ is modeled by a neural network with parameters $\psi$, s.t. $f_\psi(\boldsymbol{s}) \in \mathbb{R}^m$. The second part is the $l2$ regularization loss with all the parameters of the network lumped to $\psi$. The utility function $u_\psi$ can be expressed as follows:

$$u_\psi(\boldsymbol{s}) = ||f_\psi(\boldsymbol{s}) - \boldsymbol{c}||_2^2 \tag{4}$$

A lower value of $u_\psi(\boldsymbol{s})$ indicates that the state has been seen in $\mathcal{D}$. We modify the potential function $\Phi_\phi(\boldsymbol{s})$ and thus the extrinsic reward function, to incorporate the utility score as follows:

$$\Phi_{\phi,\psi}(\boldsymbol{s}) = \mathbf{1}\{u_\psi(\boldsymbol{s}) \leq \delta\} * \arg\max_{j \in G} \pi_\phi(g = j|\boldsymbol{s}),$$
$$r'(\boldsymbol{s}, a, \boldsymbol{s}') = \gamma\Phi_{\phi,\psi}(\boldsymbol{s}') - \Phi_{\phi,\psi}(\boldsymbol{s}), \tag{5}$$

where $\Phi_{\phi,\psi}$ denotes the modified potential function. It may be noted that as the extrinsic reward function is still a potential-based function [13], the optimality conditions between the MDP $\mathcal{M}$ and $\mathcal{M}'$ still hold as discussed previously.

---

**Algorithm 1** Learning Sub-Goal Prediction

---

  **Input:** Expert trajectory set $\mathcal{D}$
  **Output:** Sub-goal predictor $\pi_\phi(g|\boldsymbol{s})$
  $k \leftarrow 0$
  Obtain $g^k$ for each $\boldsymbol{s} \in \mathcal{D}$ using Eqn. 2
  **repeat**
    Optimize Eqn. 1 to obtain $\pi_\phi^k$
    Predict p.m.f of $G$ for each $\boldsymbol{s} \in \mathcal{D}$ using $\pi_\phi^k$
    Obtain new sub-goals $g^{k+1}$ using the p.m.f in DTW
    done = True, if $|g^k - g^{k+1}| < \epsilon$, else False
    $k \leftarrow k + 1$
  **until** done is True

---

**Supervised Pre-Training.** We first pre-train the policy network using the trajectories $\mathcal{D}$ (details in supplementary). The performance of the pre-trained policy network is generally quite poor and is upper bounded by the expert performance from which the trajectories are drawn. We then employ RL, which starts from the pre-trained policy, to learn from the subgoal based reward function. Unlike standard imitation learning algorithms, e.g., DAgger, which finetune the pre-trained policy with the expert in the loop, our algorithm only uses the initial set of expert trajectories and does not invoke the expert otherwise.

## 4 Experiments

In this section, we perform experimental evaluation of the proposed method of learning from trajectories and compare it with other state-of-the-art methods. We also perform ablation of different modules of our framework.

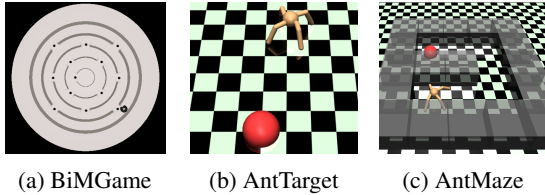

(a) BiMGame  (b) AntTarget  (c) AntMaze

Figure 2: This figure presents the three environments used in this paper - (a) Ball-in-Maze Game (BiMGame) (b) Ant locomotion in an open environment with an end goal (AntTarget) (c) Ant locomotion in a maze with an end goal (AntMaze)

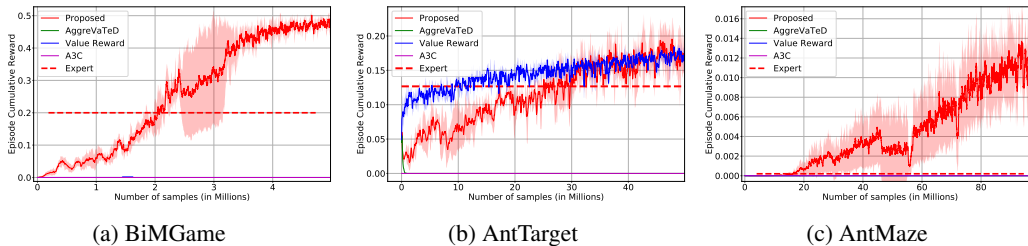

(a) BiMGame  (b) AntTarget  (c) AntMaze

Figure 3: This figure shows the comparison of our proposed method with the baselines. Some lines may not be visible as they overlap. For tasks (a) and (c) our method clearly outperforms others. For task (b), although value reward initially performs better, our method eventually achieves the same performance. For a fair comparison, we do not use the out-of-set augmentation to generate this plot.

**Tasks.** We perform experiments on three challenging environments as shown in Fig. 2. First is Ball-in-Maze Game (BiMGame) introduced in [43], where the task is to move a ball from the outermost to the innermost ring using a set of five discrete actions - clock-wise and anti-clockwise rotation by $1°$ along the two principal dimensions of the board and "no-op" where the current orientation of the board is maintained. The states are images of size $84 \times 84$. The second environment is AntTarget which involves the Ant [44]. The task is to reach the center of a circle of radius 5m with the Ant being initialized on a $45°$ arc of the circle. The state and action are continuous with $41$ and $8$ dimensions respectively. The third environment, AntMaze, uses the same Ant, but in a U-shaped maze used in [35]. The Ant is initialized on one end of the maze with the goal being the other end indicated as red in Fig. 2c. Details about the network architectures we use for $\pi_\theta$, $\pi_\phi$ and $f_\psi(s)$ can be found in the supplementary material.

**Reward.** For all tasks, we use sparse terminal-only reward, i.e., $+1$ only after reaching the goal state and $0$ otherwise. Standard RL methods such as A3C [45] are not able to solve these tasks with such sparse rewards.

**Trajectory Generation.** We generate trajectories from A3C [45] policies trained with dense reward, which we do not use in any other experiments. We also generate sub-optimal trajectories for BiMGame and AntMaze. To do so for BiMGame, we use the simulator via Model Predictive Control (MPC) as in [46] (details in the supplementary). For AntMaze, we generate sub-optimal trajectories from an A3C policy stopped much before convergence. We generate around $400$ trajectories for BiMGame and AntMaze, and $250$ for AntTarget. As we generate two separate sets of trajectories for BiMGame and AntTarget, we use the sub-optimal set for all experiments, unless otherwise mentioned.

**Baselines.** We primarily compare our method with RL methods which utilize trajectory or expert information - AggreVaTeD [8] and value based reward shaping [13], equivalent to the $K = \infty$ in THOR [14]. For these methods, we use $\mathcal{D}$ to fit a value function to the sparse terminal-only reward of the original MDP $\mathcal{M}$ and use it as the expert value function. We also compare with standard A3C, but pre-trained using $\mathcal{D}$. It may be noted that we pre-train all the methods using the trajectory set to have a fair comparison. We report results with mean cumulative reward and $\pm\sigma$ over 3 independent runs.

**Comparison with Baselines.** First, we compare our method with other baselines in Fig 3. Note that as out-of-set augmentation using $u_\psi$ can be applied for other methods which learn from trajectories, such as value-based reward shaping, we present the results for comparison with baselines without using $u_\psi$, i.e., Eqn. 3. Later, we perform an ablation study with and without using $u_\psi$. As may be observed, none of the baselines show any sign of learning for the tasks, except for ValueReward, which performs comparably with the proposed method for AntTarget only. Our method, on the other hand, is able to learn and solve the tasks consistently over multiple runs. The expert cumulative

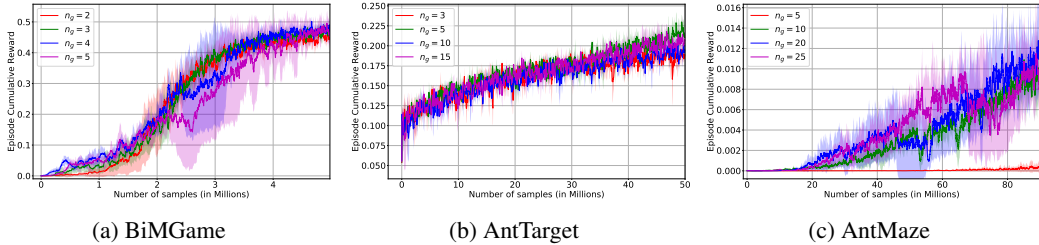

| (a) BiMGame | (b) AntTarget | (c) AntMaze |

Figure 4: (a) This plot presents the learning curves associated with different number of learned sub-goals for the three tasks. For BiMGame and AntTarget, the number of sub-goals hardly matters. However, due to the inherently longer length of the task for AntMaze, lower number of sub-goals such as $n_g = 5$ perform much worse than with higher $n_g$.

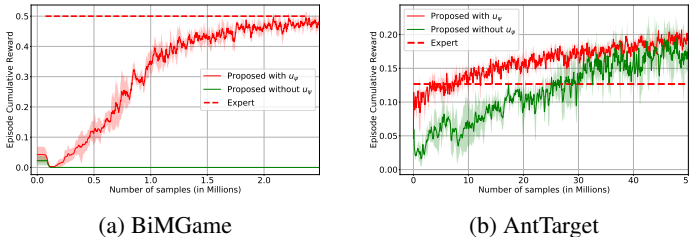

| (a) BiMGame | (b) AntTarget |

Figure 5: This plot presents the comparison of our proposed method for with and without using the one-class classification method for out-of-set augmentation.

rewards are also drawn as straight lines in the plots and imitation learning methods like DAgger [7] can only reach that mark. Our method is able to surpass the expert for all the tasks. In fact, for AntMaze, even with a rather sub-optimal expert (an average cumulative reward of only $0.0002$), our algorithm achieves about $0.012$ cumulative reward at 100 million steps.

The poor performance of the ValueReward and AggreVaTeD can be attributed to the imperfect value function learned with a limited number of trajectories. Specifically, with an increase in the trajectory length, the variations in cumulative reward in the initial set of states are quite high. This introduces a considerable amount of error in the estimated value function in the initial states, which in turn traps the agent in some local optima when such value functions are used to guide the learning process.

**Variations in Sub-Goals.** The number of sub-goals $n_g$ is specified by the user, based on domain knowledge. For example, in the BiMGame, the task has four bottle-necks, which are states to be visited to complete the task and they can be considered as sub-goals. We perform experiments with different sub-goals and present the plots in Fig. 4. It may be observed that for BiMGame and AntTarget, our method performs well over a large variety of sub-goals. On the other hand for AntMaze, as the length of the task is much longer than AntTarget (12m vs 5m), $n_g \geq 10$ learn much faster than $n_g = 5$, as higher number of sub-goals provides more frequent rewards. Note that the variations in speed of learning with number of sub-goals is also dependent on the number of expert trajectories. If the pre-training is good, then less frequent sub-goals might work fine, whereas if we have a small number of expert trajectories, the RL agent may need more frequent reward (see the supplementary material for more experiments).

**Effect of Out-of-Set Augmentation.** The set $\mathcal{D}$ may not cover the entire state-space. To deal with this situation we developed the extrinsic reward function in Eqn. 5 using $u_\psi$. To evaluate its effectiveness we execute our algorithm using Eqn. 3 and Eqn. 5, and show the results in Fig. 5, with legends showing without and with $u_\psi$ respectively. For BiMGame, we used the optimal A3C trajectories, for this evaluation. This is because, using MPC trajectories with Eqn. 3 can still solve the task with similar reward plots, since MPC trajectories visit a lot more states due to its short-tem planning. The (optimal) A3C trajectories on the other hand, rarely visit some states, due to its long-term planning. In this case, using Eqn. 3 actually traps the agents to a local optimum (in the outermost ring), whereas using $u_\psi$ as in Eqn. 5, learns to solve the task consistently (Fig. 5a).

For AntTarget in Fig. 5b, using $u_\psi$ performs better than without using $u_\psi$ (and also surpasses Value based Reward Shaping). This is because the trajectories only span a small sector of the circle (Fig. 7b) while the Ant is allowed to visit states outside of it in the RL step. Thus, $u_\psi$ avoids incorrect sub-goal assignments to states not well-represented in $\mathcal{D}$ and helps in the overall learning.

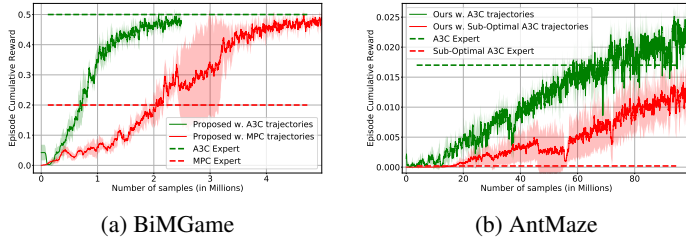

(a) BiMGame             (b) AntMaze

Figure 6: This plot presents a comparison of our proposed method for two different types of expert trajectories. The corresponding expert rewards are also plotted as horizontal lines.

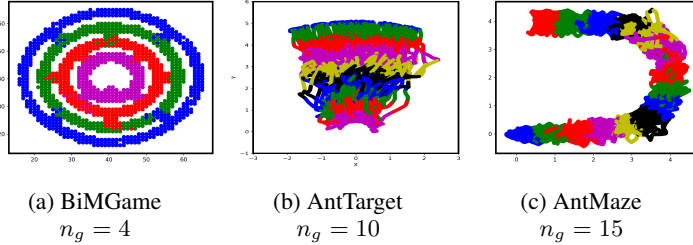

(a) BiMGame
$n_g = 4$

(b) AntTarget
$n_g = 10$

(c) AntMaze
$n_g = 15$

Figure 7: This figure presents the learned sub-goals for the three tasks which are color coded. Note that for (b) and (c), multiple sub-goals are assigned the same color, but they can be distinguished by their spatial locations.

**Effect of Sub-Optimal Expert.** In general, the optimality of the expert may have an effect on performance. The comparison of our algorithm with optimal vs. sub-optimal expert trajectories are shown in Fig. 6. As may be observed, the learning curve for both the tasks is better for the optimal expert trajectories. However, in spite of using such sub-optimal experts, our method is able to surpass and perform much better than the experts. We also see that our method performs better than even the optimal expert (as it is only *optimal* w.r.t. some cost function) used in AntMaze.

**Visualization.** We visualize the sub-goals discovered by our algorithm and plot it on the x-y plane in Fig. 7. As can be seen in BiMGame, with 4 sub-goals, our method is able to discover the bottle-neck regions of the board as different sub-goals. For AntTarget and AntMaze, the path to the goal is more or less equally divided into sub-goals. This shows that our method of sub-goal discovery can work for both environments with and without bottle-neck regions. (See supplementary for more visualizations).

## 5 Discussions

The experimental analysis we presented in the previous section contain the following key observations:
- Our method for sub-goal discovery works both for tasks with inherent bottlenecks (e.g. BiMGame) and for tasks without any bottlenecks (e.g. AntTarget and AntMaze), but with temporal orderings between groups of states in the expert trajectories, which is the case for many applications.
- Experiments show, that our assumption on the temporal ordering of groups of states in expert trajectories is soft, and determines the granularity of the discovered sub-goals (see supplementary).
- Discrete rewards using sub-goals performs much better than value function based continuous rewards. Moreover, value functions learned from long and limited number of trajectories may be erroneous, whereas segmenting the trajectories based on temporal ordering may still work well.
- As the expert trajectories may not cover the entire state-space regions the agent visits during exploration in the RL step, augmenting the sub-goal based reward function using out-of-set augmentation performs better compared to not using it.

## 6 Conclusion

In this paper, we presented a framework to utilize the demonstration trajectories in an efficient manner by discovering sub-goals, which are waypoints that need to be completed in order to achieve a certain complex goal-oriented task. We use these sub-goals to augment the reward function of the task, without affecting the optimality of the learned policy. Experiments on three complex task show that unlike state-of-the-art RL, IL or methods which combines them, our method is able to solve the tasks consistently. We also show that our method is able to perform much better than sub-optimal experts used to obtain the expert trajectories and at least as good as the optimal experts. Our future work will concentrate on extending our method for repetitive non-goal oriented tasks.

**Acknowledgement.** This work was partially supported by US NSF grant 1724341 and Mitsubishi Electric Research Labs.

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
