[Supplementary Material · 4556-supp-CameraReady.pdf]

# Supplementary for Learning from Trajectories via Subgoal Discovery

**Sujoy Paul**[1]
supaul@ece.ucr.edu

**Jeroen van Baar**[2]
jeroen@merl.com

**Amit K. Roy-Chowdhury**[1]
amitrc@ece.ucr.edu

[1]University of California-Riverside    [2]Mitsubishi Electric Research Laboratories (MERL)

## Contents

## 1   Variations in Sub-goals and Assumptions on Trajectories

In this section, we show visualizations of the learned sub-goals for different number of sub-goals. Fig. 1 and Fig. 2 shows the visualizations for the AntMaze task using sub-optimal and optimal trajectories respectively. Fig. 3 and Fig. 4 shows the visualizations for BiMGame and AntTarget respectively. It may be observed in Fig. 1, that with high $n_g$, although our algorithm starts from the specified number of sub-goals, at the end of the sub-goal learning process, it ends up discovering fewer sub-goals (shown in brackets), $25 \rightarrow 21$ and $20 \rightarrow 18$. However, with optimal trajectories (Fig. 2), our algorithm is able to discover the pre-specified number of sub-goals (at least till $n_g = 30$). This is due to the fact that the variations in the path taken by the optimal trajectories are much less than the sub-optimal trajectories. Thus, our algorithm is able to cluster the states more appropriately for optimal than sub-optimal trajectories. This actually shows the claim we make in the paper, that our assumption that certain groups of states should follow some temporal ordering in the trajectories, are only soft and the degree by which they deviate determine the number and thus the granularity of the discovered sub-goals. Moreover, as we see in Fig. 4c of the paper, even with sub-optimal trajectories, a low number of pre-specified sub-goals (such as $n_g = 10$) performs almost as good as with pre-specified $n_g = 25$, which actually discovers 21 sub-goals.

## 2   Variations with Number of Trajectories

In this section, we evaluate the performance of the proposed method with changes in the number of trajectories. Specifically we use our method with 125 and 250 trajectories for the AntTarget environment and plot them in Fig. 5a. As can be observed from the plot that the difference in performance for with and without using $u_\psi$ is more with fewer number of trajectories. We also show the performance of our algorithm with variations in number of trajectories and number of sub-goals in Fig. 5b. Fig. 5c and 5d show that our method to learn sub-goals is data-efficient as the sub-goals learned using 125 trajectories are similar to that learned with 250 trajectories. So, the small increment

(a) $(5, 5)$    (b) $(10, 10)$    (c) $(15, 15)$    (d) $(20, 18)$    (e) $(25, 21)$

Figure 1: This figure presents the visualizations of the discovered sub-goals for AntMaze using the **sub-optimal** set of expert trajectories with different number of pre-specified sub-goals ($n_g$). The values as caption denote (no. of pre-specified sub-goals, no. of sub-goals learned).

(a) $(5, 5)$    (b) $(10, 10)$    (c) $(20, 20)$    (d) $(25, 25)$    (e) $(30, 30)$

Figure 2: This figure presents the visualizations of the discovered sub-goals for AntMaze using the **optimal** set of expert trajectories with different number of sub-goals ($n_g$) as input. The values as caption denote (no. of pre-specified sub-goals, no. of sub-goals learned).

(a) $(2, 2)$    (b) $(3, 3)$    (c) $(4, 4)$    (d) $(5, 5)$

Figure 3: This figure presents the visualizations of the discovered sub-goals for BiMGame with different number of sub-goals ($n_g$) as input. The values as caption denote (no. of pre-specified sub-goals, no. of sub-goals learned).

(a) $(5, 5)$    (b) $(10, 10)$    (c) $(15, 15)$    (d) $(20, 20)$    (e) $(25, 24)$

Figure 4: This figure presents the visualizations of the discovered sub-goals for AntTarget using the expert trajectories with different number of sub-goals ($n_g$) as input. The values as caption denote (no. of pre-specified sub-goals, no. of sub-goals learned).

in the performance when using 250 vs. 125 trajectories is probably due to better pre-training of the network with more variations in trajectories.

## 3   Generating sub-optimal trajectories for BiMGame

In this method of generating trajectories for the BiMGame environment, we leverage the internal physics engine of the simulator to forward propagate the state in time and generate trajectories by optimizing the cumulative reward function in an Model Predictive Control (MPC) manner. Formally,

(a)

(b)

(c) 125 trajectories, $n_g = 5$

(d) 250 trajectories, $n_g = 5$

Figure 5: (a) This figure shows the performance on AntTarget environment with variations in the number of trajectories used for learning. The plot also shows the performance with and without using out-of-set augmentation. The number of sub-goals used is $n_g = 5$. (b) This plot shows performance with different combination of number of sub-goals ($n_g$) and number of trajectories used for training. (c) and (d) vsisualizes the sub-goals learned for $n_g = 5$ with 125 and 250 trajectories.

at time step $t$, we obtain the optimal action set $a^*_{t:t+H-1}$ from $t$ to $t+H-1$ by solving the following:

$$\underset{a_{t:t+H-1}}{\arg\max} \sum_{t'=t}^{t+H-1} r(s_{t'}, a_{t'}, s_{t'+1}), \text{ s.t., } s_{t'+1} = \mathcal{S}(s_{t'}, a_{t'}),\qquad(1)$$

where $\mathcal{S}$ is the simulator, $r(s_t, a_t, s_{t+1}) = d(s_{t+1}) - d(s_t)$ is the reward, $d(s_t)$ is the radial distance of the ball at time $t$ from the center of the board, $H$ is the horizon of optimization and $a_{t:t+H-1}$ is a set of actions. We only take the first action $a^*_t$, move to state $s_{t+1}$ and repeat Eqn. 1. As we use a non-differentiable simulator, we employ a random shooting strategy [1] where we sample $K$ sets of $a_{t:t+H-1}$ and choose the one which maximizes the rewards. We use $K, H = 10$ empirically. Note that the reward and the random shooting may not lead to the shortest path, thus making the trajectories sub-optimal.

## 4 Supervised Pre-Training

As discussed previously, an initial way to utilize the trajectories is by pre-training the policy network $\pi_\theta$ using the trajectory set $\mathcal{D}$ in a supervised learning framework. We pre-train the network by optimizing the following:

$$\theta^* = \underset{\theta}{\arg\min} \sum_{i=1}^{n_d} \sum_{t=1}^{n_i} l(\pi_\theta(a|s_{ti}), \boldsymbol{a}^*_{ti}) + \lambda||\theta||^2_F\qquad(2)$$

where $l$ is the loss function which can be cross-entropy or regression loss depending on discrete or continuous actions. Note that for continuous actions in AntTarget and AntMaze, the action variables comprise of $(\mu, \sigma)$. The second part of Eqn. 2 is the $l2$ regularization loss. The policy obtained after

optimizing Eqn. 2 possesses the ability to take actions with low error rates at the states sampled from the distribution induced by the trajectory set $\mathcal{D}$. However, a small error at the beginning would compound quadratically [2] with time as the agent starts visiting states which are not sampled from the distribution of $\mathcal{D}$. Algorithms like DAgger can be used to fine-tune the policy by querying expert actions at states visited after executing the learned policy. This query to the expert is often very costly and even may not be feasible in some applications. More importantly, as DAgger aims to mimick the expert, it can only reach its performance and not better than that. For this reason, we fine-tune the policy using RL with the extrinsic reward function obtained after identifying the sub-goals.

## 5   Network Architectures

We follow the architecture of A3C [3] and share parameters between the policy and the state value estimation network. To model $\pi_\theta$ in BiMGame, we use a CNN with architecture Conv-Conv-FC-RNN followed by two heads: one for policy network and another for state value estimation. We append the previous step action as additional input to the RNN step [4]. To model $\pi_\theta$ for AntTarget and AntMaze, we use the architecture FC-FC-FC-RNN, again followed by two heads for policy and state value estimation. For the policy part, we predict the mean and standard deviation. We use similar architectures for the respective tasks for $\pi_\phi$ and $f_\psi(s)$ with modifications in the final layer to suit their purpose. We do not use a RNN for $\pi_\phi$ or $f_\psi(s)$.

## 6   Effect of Number of Sub-goals and Number of Trajectories

Fig. 4a and 4b (of paper) show that the performance remains similar for $n_g \geq 2$. Tasks cannot be solved with $n_g = 1$, i.e., all the states clustered to a single subgoal, and thus no rewards from subgoals. However, we do see that for AntMaze Fig 4c (of paper) the performance is dependent on the number of subgoals, due to the longer time horizon required to solve the task. We also performed an experiment on BiMGame with fewer trajectories (250 instead of 400). With fewer trajectories, pre-training performance is lower and we will therefore need more steps in RL, with more frequent rewards. This can be seen in Fig. 6 which shows that for 250 trajectories, $n_g = 2$ is not able to solve the task, but $n_g = 4$ is able to solve it.

Figure 6: Effect of number of sub-goals and trajectories on BiMGame.

## 7   Analysis of Baselines Performance

In our experiments, we observe that the baselines cannot solve BiMGame & AntMaze even with pre-training using the optimal trajectories. AggreVaTeD in particular has a strong dependency on the quality of the value function [5]. For value reward, we observe some progress in BiMGame and AntMaze towards the goal. For BiMGame it is able to get into the $2^{nd}$ & $3^{rd}$ ring, and for AntMaze it is able to reach the first turn. Fig. 7a, 7b shows this as the distribution of visited states (sampled regularly). Nevertheless, their cumulative terminal-only reward is 0. We see similar trends for AggreVaTeD.

(a) BiMGame

(b) AntMaze

Figure 7: Distribution of visited states for Value-Based Rewards (VBRS) for BiMGame and AntMaze. Brighter color means visited more number of times.