[Reviews · NeurIPS 2019]

Reviewer 1



The paper describes a framework to learn from different expert trajectories how to decompose a task into smaller sub-goals. This is used to define a reward function and imitation learning is first used to learn a policy followed by reinforcement learning given the sub-goals. Also a module to deal with new states based on one-class learning is used to provide robustness to the system. A nice feature of the system is that it can learn tasks even with sub-optimal trajectories. One limitation is that the system assumes a sequential order of the sub-goals and all the trajectories must start from the same initial state distribution. This means that the system is unable to deal with sequences following different paths or from different initial states. The number of partitions is given in advance, and although the number seems not to be too relevant, in some cases it may hamper the performance of the system. It would be great if there could be some guidance or process to make this selection more easily. The DTW process is not sufficiently clarified. The system evaluates DTW for all pairs of trajectories? It is also not clear how the goal assignment is performed for each state after DTW. It seems to be an implicit assumption that all the trajectories must have very similar number of states, is that the case? It is clear that pre-learning process helps the subsequent RL process, however, it is not clear why the information from sub-goals cannot be used from the beginning with only one learning process. In the experiments the trajectories given to the system are not very good, not surprisingly the other methods perform poorly, however, with information from sub-goals, this initial guidance proves to deal even with noisy information. I will be very useful to see how the other methods perform with close-to-optimal trajectories. After reading the author's rebuttal, I have chosen to maintain my score for these reasons: I believe that automatically decomposing a problem and being able to use information from noisy trajectories are relevant topics. Although there are some concerns with this paper, and I will not argue too much for its acceptance, I still think that it opens some new lines of research for solving more complex tasks.

Reviewer 2



Summary: The objective of this paper is to produce an RLfD algorithm that can both increase learning speed vs. sparse-rewards from scratch, and exceed expert performance. The main assumption is that all expert trajectories go through the same sequence of abstract steps, which are internal to the expert and implicit in the demos. By training classifiers to segment these trajectories into a sequence of monotonically increasing "goals", the goals can be used as sparse shaping cues. Abstracting over states in this fashion prevents the shaping signal from fitting the idiosyncrasies of the expert, and allows the experimenter to control the granularity of expert's state-density distribution to follow during training. The main contribution is an EM-style algorithm for learning goal labels from expert trajectories. It works by alternating between (a) assigning a goal-label to all states in the dataset such that goals are monotonically increasing over the demos, and (b) fitting an n-way classifier to maximize log-prob of the labels. The authors also use an explicit anomaly detector to avoid shaping when far from the expert data, which showed a positive effect in their 2 most challenging tasks. Comments: "252 The poor performance of the ValueReward and AggreVaTeD can be attributed to the imperfect value 253 function learned with a limited number of trajectories. Specifically, with an increase in the trajectory 254 length, the variations in cumulative reward in the initial set of states are quite high. This introduces a 255 considerable amount of error in the estimated value function in the initial states, which in turn traps 256 the agent in some local optima when such value functions are used to guide the learning process." > This argument should apply to the proposed method as well. The proposed method is effectively learning a value-like-function on a discretized state-space, and using it in the same potential-based way the regular value function would be. How do the authors explain the robustness to local-optima vs. the baseline methods? These tasks are not especially complex, and I find it suspicious that the authors were unable to get VBRS and aggrevated to work at all. "285 We also see that our method performs better than even the optimal expert (as it is only optimal w.r.t. some cost function) used in AntMaze" > Aren't you evaluating on that same cost function? If so it sounds like an under-fitting issue, and if not the difference in cost function should be discussed. Fig 3) all baselines fail utterly in tasks 1 & 3, which is suspicious . See comment below Fig 4a) why does performance drop as nd 2->5? Why not show n=1? It seems that shaping may not be needed for this task. Fig 4b) no effect of ng -> suggests shaping isn't really necessary here and that the main effect comes from pretraining. Both 4a and 4b makes a weak case for the proposed method, since the goal sparsity has little effect on learning speed or final performance. "Although the equipartition labels in Eqn. 2 may have a many-to-one mapping from state to sub-goals, the learned network modeling πφ provides a one-to-one mapping." > How can a mapping from Rn -> Nk be one-to-one? Would be interesting to see if expert performance can be exceeded when training solely from expert demos, ie without any environment reward at all. There’s reason to believe this is possible, but it wasn’t discussed explicitly Final comment: The main approach to goal-labeling works by iteratively assigns labels to states in ascending order over trajectories, and training a classifier to minimize a cross-entropy loss with respect to these labels. It seems the root issues are the need to preserve sequential ordering of class-labels over time, and the need to preserve the balance of goal-labels across the dataset. The narrative and algorithm might benefit from expressing these criteria more explicitly, e.g. by using Ordinal-Regression to capture the sequentiality of discrete labels by construction in the model, and using a prior (e.g. uniform) or KL-regularizer on labels to preserve entropy. This comment is merely a suggestion, and did not affect the review.

Reviewer 3



The paper proposes an algorithm to solve sparse reward MDPs with expert demonstrations by learning a set of sub-goals from the demonstrations. To learn the sub-goals, they train a multi-class classification model that assigns agent states to different clusters. To ensure that states later in the trajectory do not belong to a previous sub goal, they used dynamic time warping to adjust the output of the model and use it to generate new training targets. To deal with the problem that the demonstrations do not cover the entire state space, they used the deep one class classification algorithm so that the final algorithm only uses prediction for states near the demonstration trajectories. The method seems reasonable to me and the results seems solid. The paper is not difficult to follow. However, I have a few concerns and questions regarding the proposed approach, as listed below: First, it’s not clear from the text how important it is to learn a neural network model for the sub-goals. From the results in Figure 7, it seems that the results could be produced by some clustering algorithm as well? For example, what would happen if one uses the equipartition (eq 2) with a nearest neighbor classifier to specify the sub-goals? Also, despite that the use of DTW to ensure temporal consistency is interesting, it’s not verified if it’s necessary to do so in the experiments. Also, the sub-goals used in the experiments seems to be the COM position of the robots (based on the look of Figure 7) and all the problems have a navigation nature. It’s not very clear how well can this generalize to higher dimensional states, like for manipulation or locomotion tasks where the full joint state of the robot is needed. What happens if the expert trajectories solve the problem in different ways, i.e. there are branches in the middle of the trajectory? In addition, there has been some existing work that uses self-imitation to deal with sparse reward problems [1], which could be applied to the problem of interest here. It would be nice if some comparisons can be done. [1]. Learning Self-Imitating Diverse Policies. Gangwani et al. ICLR 2019. ================================================== The authors' response has addressed most of my concerns and the additional experiments regarding sub-goal modeling is well appreciated. I have updated my score.

[Author Response · NeurIPS 2019]

We sincerely thank the reviewers for their helpful comments.

**Baselines.** The baselines do not solve BiMGame & AntMaze even with optimal trajectories. However, we observe
that for BiMGame it is able to get into the $2^{nd}$ & $3^{rd}$ ring, and in AntMaze, near the first turn. Fig. D, E shows this as
the distribution of visited states (sampled regularly) for value reward (VBRS). We see similar trends for AggreVaTeD.
Although they stagnate after making some progress, their cumulative terminal-only reward is $0$. (see Line 300-302).

**Reviewer 1:** • **1a. Order:** We only assume ordering of state groups, which is implicit in many tasks. The trajectories
may bifurcate to take different paths to the goal, (as in BiMGame), but our method is able to efficiently learn the
subgoals. We empirically show in suppl. that the order assumption is soft, and not strict. Also, it is hardly a limitation
to assume the trajectories to start from initial states, as it do not incur an extra cost, and often followed in literature.
• **1b. No. of subgoals:** It is a hyper-parameter which we can decide from task info. (Line 257-259) or tuning methods.
• **1c. DTW & length:** To assign one of the $n_g$ subgoals to a trajectory state, we perform DTW between subgoals,
represented as a sequence of $n_g$ one-hot vectors, and the subgoal pred. p.m.f. of trajectory states (Line 158-160). We do
not assume the trajectories to have similar length, for e.g., it ranges in $[53, 850]$ for BiMGame, $[580, 2470]$ for AntMaze.
• **1d. One learning process:** Subgoals are only required in the RL step and not to pre-train. One can start directly from
RL with subgoals, without pre-training, which will still work, but with many more samples. One can inject the expert
trajectories in an off-policy RL method, but it is non-trivial to schedule the sampling of these sub-optimal trajectories.
• **1e. Trajectories in other methods:** As the motivation is to show that our method can learn from sub-optimal
trajectories, for a fair comparison, we use the same sub-optimal trajectories to generate the sub-goals in our method, to
learn the value function in the baselines as well as pre-training the policies in all methods. See "Baselines" above.

**Reviewer 2:** • **2a. Perf. of AggreVaTeD & VBRS:** Although subgoal rewards have a similar form as VBRS, the
subgoal relabeling step via prediciton+correction helps our method to efficiently learn the subgoals. Fig. B show that
without correction using DTW, the subgoals learned are noisy and we find that it cannot learn the task. Also, the strong
dependency of AggreVaTeD on the quality of the value func. is discussed in [9, 14]. See "Baselines" & Fig. D, E.
• **2b. Different cost func.:** As our expert is a model trained with dense rewards with 'A3C' (Line 228), it is optimal
w.r.t. to the dense rewards, but may not be optimal w.r.t. the sparse terminal-only rewards, which we plot in the figures.
• **2c. Effect of $n_g$, necessity of shaping, and $n_g = 1$:** Although Fig. 4a,b show that $n_g >= 2$ works for BiMGame &
AntTarget, it does not imply that shaping is not necessary for these tasks. This is because, for $n_g = 1$ (equivalent to
A3C in Fig. 3), all states are grouped into a single set, thus no rewards from subgoals, and it fails to learn even with
pre-training. Also, Fig. C show that with lesser demos, as pre-training is not as good, $n_g = 2$ fails, but $n_g = 4$ works.
• **2d. Fig. 3:** See 2a. • **2e. Fig. 4a:** The drop in the middle for $n_g = 5$ is due to only one of the random RL runs not
reaching the terminal for some iter. It is an outlier as the other runs are similar to the other $n_g$. • **2f. Fig. 4b:** See 2c.
• **2g. Mapping.** We meant that in the initial equipartition (Eqn. 2), exactly same states from different demos may have
different labels. However, the learned network $\pi_\phi$ will always have unique subgoals for each state. We will rephrase.
• **2h. Exceeding expert perf.:** Supervised pre-training with expert demos can only achieve the expert perf., albeit with
a lot of demos, but cannot surpass the expert [7, 14]. E.g., in BiMGame, we found that models pre-trained with $1500$ &
$3000$ expert demos (of perf. $0.2$), perform $0.147$ & $0.151$ respectively. Our method perf. $0.5$ after RL with subgoals.
• **2i. Final comment:** Thanks for the great suggestion. • **2j. Non-trivial tasks:** BiMGame and AntTarget are
non-trivial tasks as it fails without reward shaping. See 2c. • **2k.:** $n_g$ is inherently related to the task horizon, e.g.,
$n_g = 4$ works for BiMGame, but not as fast for AntMaze. • **2l. Human-in-loop:** While we do not ask a human for the
expert demos (common in literature [8, 9, 14]), the MPC demos can be considered to be similar to human-generated,
due to its method of forward simulation in time and selecting the best action.

**Reviewer 3:** • **3a. Use of Neural Networks (NN):** We use NNs to learn the subgoals as the inputs are high dimensional
(Line 218, 220) and NN learns the subgoal prediction end-to-end. As suggested, we predict the subgoals using nearest
neighbors in high dim. (for BiMGame) and visualize in Fig. A. We also tested our method without DTW and visualize
in Fig. B. Both figures are quite noisy compared to our method and using them with RL is not able to learn the task.
• **3b. High-dimensional states:** We do learn from high dim. states. The states are $84 \times 84$ dim. images in BiMGame
and 41 dim. robot state in AntTarget & AntMaze (Lines 218, 220). But, to visualize, we plot the 2D x-y coordinates.
• **3c. Trajectory branching:** We do see such trajectories in BiMGame, as there are multiple paths to the goal and
trajectories can bifurcate to reach the goal. Results show that our method can learn from such trajectories as well.
• **3d. Self-imitation compare:** These works utilize its past experiences to learn faster. However, as it do not use expert
demos, to be fair, we do not compare with them and focused on evaluating methods which use expert demos, but we
can add this comparison in supplementary.

(A) Using kNN    (B) No DTW    (C) BiMGame    (D) VBRS-BiMGame    (E) VBRS-AntMaze

[Meta-Review · NeurIPS 2019]

The reviewers felt that the paper is an interesting approach to learning subgoals via demonstration trajectories. The reviewers also felt that the rebuttal addressed many of their concerns. The paper would be improved by adding additional experiments and analysis, such as the analysis shown in the rebuttal.